# Effects of Glutamine Starvation on SHVV Replication by Quantitative Proteomics Analysis

**Junlin Liu** [1], **Yulei Zhang** [2], **Xiaoyan Liu** [1], **Hantao Zhang** [1], **Yi Liu** [3], **Keping Chen** [1], **Min Tang** [1] and **Lindan Sun** [1,*]

1   School of Life Sciences, Jiangsu University, Zhenjiang 212013, China
2   Guangdong South China Sea Key Laboratory of Aquaculture for Aquatic Economic Animals, Guangdong Ocean University, Zhanjiang 524088, China
3   Jiangsu Academy of Agricultural Sciences, Nanjing 210014, China
*   Correspondence: sunlindan@ujs.edu.cn

**Abstract:** Snakehead vesiculovirus (SHVV), a strain of negative-stranded RNA viruses extracted from sick snakehead fish (*Ophicephalus striatus*), may pose a threat to the health of snakehead fish. Previous research has proved that the replication of SHVV can be significantly inhibited by glutamine starvation. To study how glutamine starvation inhibits SHVV replication, channel catfish ovary (CCO) cells with SHVV cultivated in the glutamine-free medium or the complete medium were used to investigate the differentially expressed proteins (DEPs). The results showed that 124 up-regulated and 246 down-regulated proteins were involved in many viral replication physiological processes, such as autophagy, post-translational modifications machinery, and functional pathways, including the PI3K-Akt signaling pathway and mTOR signaling pathway. Furthermore, a few proteins, such as Akt and Hsp90, which have been confirmed to be involved in the replication of RNA viruses, were also significantly differentially expressed. Taken together, our study demonstrated that glutamine starvation affects various functional pathways and the expression of some key proteins related to RNA viral replication, which will benefit future studies on the replication mechanisms of SHVV and the prevention of SHVV infection.

**Keywords:** SHVV; proteomics; pathogenesis; RNA viral replication

## 1. Introduction

Viruses target host cell mechanisms to promote their efficient replication, and various physiological processes have been proven to participate in RNA virus replication. During infection, the virus hijacks the host post-translational modifications machinery (PTMs), including ubiquitination, acetylation, SUMOylation, methylation, phosphorylation, and glycosylation to regulate its replication [1–4]. Autophagy is a highly conserved cellular mechanism [5], and some biomolecules inside the cells including misfolded proteins [6] are delivered to lysosomes and degraded there [7], participating in the cell protective effect [8]. It has been reported that autophagy induced by negative-strand RNA viruses, such as the influenza A virus [9,10], viral hemorrhagic septicemia virus (VHSV) [11], infectious salmon anemia virus (ISAV) [12], Sendai virus (SeV) [13], simian virus 5 (SV5) [14], and others, plays an essential role in regulating their replication. Our earlier study has found that glutamine starvation inhibits SHVV replication via inducing autophagy [15]. Furthermore, various biological and physiological processes participating in viral replication have been reported to involve multiple functional pathways [13].

Glutamine has been proven to contribute significantly to the immune system [16]. It has been revealed that glutamine is also essential for the replication of RNA viruses [17,18]. Glutamine is converted to α-ketoglutaric acid via glutaminase and participates in the tricarboxylic acid (TCA) cycle to regulate RNA virus replication [19,20]. As the hub for biosynthesis, the TCA cycle provides the energy source for RNA viral replication [21,22].

It has been confirmed that glutamine starvation prevents the RNA virus from replicating itself via multiple functional pathways [23–25].

SHVV, a single negative-strand RNA virus, has caused huge losses in snakehead fish breeding in China [26]. The SHVV genome encodes the following five structural proteins: nucleoprotein (N), phosphoprotein (P), matrix protein (M), glycoprotein (G), and large polymerase (L) [27,28]. It has been shown that SHVV can infect CCO cells easily [27] and that glutamine starvation has an inhibitory effect on SHVV replication. However, the pathogenic mechanism of SHVV infection in CCO cells and the potential mechanism between glutamine starvation and SHVV replication are still enigmatic. Two experimental settings were designed to investigate these questions. CCO cells infected with SHVV cultured in a complete medium were designated as the experimental group and CCO cells without SHVV cultured in a complete medium were designated as the control group to investigate the pathogenicity of SHVV on CCO cells. CCO cells with SHVV cultured in a glutamine-free medium were designated as the experimental group and CCO cells infected with SHVV cultured in a complete medium were designated as the control group to investigate the effect of glutamine starvation on SHVV replication. The quantitative proteomic analysis of the total proteins of each group was performed. The results revealed that inhibition of SHVV replication by glutamine starvation involves multiple signaling pathways and several key proteins.

## 2. Materials and Methods

### 2.1. Virus and Cells

SHVV was extracted from sick snakehead fish culture on a farm in Guangdong Province, China, and stored at $-80\ ^{\circ}$C in our laboratory [26]. CCO cells were kindly provided by Dr. Hong Liu from Shenzhen Animal and Plant Inspection and Quarantine Technology Center (Shenzhen, China) and were amplified and maintained in a minimal essential medium (MEM, GIBCO, Carlsbad, CA, USA) with 10% fetal bovine serum (FBS, GIBCO, Carlsbad, CA, USA) at 25 $^{\circ}$C [27], with a PH of 7.2~7.4, 95% humidity and 5% $CO_2$ [29]. CCO cells were applied every 5 days after a dense cell monolayer was formed.

### 2.2. CCO Cell Culture and SHVV Infection

Dulbecco's Modified Eagle Medium (DMEM, Invitrogen) without d-glucose, l-glutamine, sodium pyruvate, and phenol red was used as a glucose- and glutamine-free medium. D-glucose (1 g/L) was added into the glucose- and glutamine-free medium to prepare a glutamine-free medium, and 2 mM L-glutamine was added to the glutamine-free medium to prepare a complete medium [30]. CCO cells were incubated with SHVV (MOI = 1) for 2 h to prepare infected cells. The CCO cells with or without SHVV were raised in the glutamine-free medium or complete medium for 24 h, based on the experimental design. Three parallel replicates were created in each experimental group.

### 2.3. Protein Extraction and Quantitation

RIPA Lysis Buffer (50 mM Tris-HCl (pH = 7.4), 150 mM NaCl, 1% NP-40, 0.5% NA-deoxycholate, 0.1% SDS, and EDTA) (Beyotime, Shanghai, China), and PMSF (1:100) (Beyotime, Shanghai, China) were used to extract total protein from CCO cells culturing in a dissimilar medium. We used a BCA kit (Thermo Fisher Scientific, Shanghai, China) for the protein concentration determination [31], following the manufacturer's program. The extracted samples were analyzed via 12% SDS-PAGE [32] (SDS: Sinopharm, Shanghai, China) for the quality control of the proteins that were subjected to the subsequent experiments.

### 2.4. Protein Alkylation and Trypsin Digestion

Protein (100 μg) was solubilized to 10 mM Tris (2-carboxyethyl) phosphine (Thermo Fisher Scientific, Shanghai, China), and incubated in the experimental systems at 37 $^{\circ}$C for 60 min. Proteins samples were reduced with 10 mmol/L DTT (Thermo Fisher Scientific, Shanghai, China) for 30 min, followed by alkylation with 55 mM iodoacetamide (Sigma,

Shanghai, China) for 60 min in the dark. Each 100 μg of protein was digested with trypsin enzyme (PROMEGA, Beijing, China) (1 mg trypsin enzyme/50 smg protein) at 37 °C overnight.

### 2.5. Label-Free Lc-Ms/MS Analysis

All treated proteins were examined by label-free liquid chromatography-tandem mass spectrometry (LC-MS/MS) for detection [33] via the nano Elute UHPLC System (Bruker Daltonics, Bremen, Germany), with one LC-MS/MS assay per sample. The analytical column specification was 25 cm × 75 μm, 2 μm, and the flow rate was set to 300 nL/min constantly. Solvent A (0.1% formic acid + 99.9% ddH$_2$O) and solvent B (80% acrylonitrile + 19.9% ddH$_2$O) were used as the mobile phase.

The data-dependent acquisition (DDA) mode was applied as the data acquisition mode [34]. The settings were as follows: the capillary voltage was set at 2.0 kV, and the capillary temperature was 300 °C. The scanning range of MS ($m/z$) was subjected at 350–1300 $m/z$, and normalized collision energies were 28%. Automatic gain control (AGC) was $1 \times 10^5$ ions, and the maximum ion implantation time was 30 ms. The DDA mode with the top 20 ions was used for secondary ion mass spectroscopy (SIMS) acquisition, the dynamic exclusion time was 40 s, and the DDA cycle time was 3 s.

### 2.6. Database Search and DEPs Screen

The resulting LC-MS/MS data were processed via the Proteome Discoverer (Thermo Fisher Scientific, MA, USA) [35] for protein identification and quantification. The Universal Protein (Uniprot) database (https://www.uniprot.org/) (accessed on 25 March 2022) was used for protein database construction [36]. The taxonomic identifier that was used for the protein search was 7998 (*Ictalurus punctatus*), and there are 43064 entries in the database now. Proteins with fold-change (FC) >2 or <0.5 and $p$-value < 0.05 were considered to be significantly differentially expressed and were screened by using the limma package in R software (4.1.2, Ross Ihaka and Robert Gentleman, The University of Auckland, Auckland, New Zealand) [37].

### 2.7. Bioinformatics Analysis

Protein functions were annotated [38] based on the Kyoto Encyclopedia of Genes and Genomes (KEGG) database (https://www.genome.jp/kegg/) (accessed on 13 April 2022), Gene Ontology (GO) database (http://geneontology.org/) (accessed on 14 April 2022), and Clusters of Orthologous Groups (COG) database (https://www.ncbi.nlm.nih.gov/research/cog-project/) (accessed on 15 April 2022). BLAST2GO (2.5.0) [39] was used for GO annotation and KOBAS (2.1.1) [40] was used for KEGG annotation. DEPs were analyzed with goatools (Tang etc. San Francisco Bay Area, USA), a library of Python (3.7.4, Python Software Foundation, DE, USA) [41] for GO pathway analysis, and Python (3.7.4, Python Software Foundation, DE, USA) [42] for KEGG pathway analysis. When the $p$-value < 0.05, the GO function was considered a significant enrichment, and the KEGG pathway was defined as a significantly enriched KEGG pathway. MultiLoc2 (KohlbacherLab, Tübingen, Germany), a tool that needs Python (3.7.4, Python Software Foundation, DE, USA) environment [43] was used for protein subcellular localization.

The STRING database (https://cn.string-db.org/) (accessed on 26 April 2022) was used for protein–protein interaction (PPI) network generation, and key nodes were obtained according to the degree centrality.

### 2.8. Statistical Analysis

$p$-values were obtained via Student's $t$-test statistical analysis to determine DEPs, and $p$-value < 0.05 was used as the criterion for determining significant differences. Fisher's exact test was applied for enrichment analysis, and $p$-value < 0.05 was identified as significant.

## 3. Results

### 3.1. Experimental Design and the Identification of Differentially Expressed Proteins (DEPs)

The experimental design was divided into the following two settings: CCO cells with or without SHVV cultured in a complete medium were used to investigate how SHVV affects CCO cells. The objective was to inquire into the effects of inhibition of glutamine starvation on SHVV replication, and CCO cells with SHVV cultured in a glutamine-free medium or a complete medium were used. Three parallel replicates were created for each experimental group.

Principal component analysis (PCA) models were used to analyze differences between groups in each experimental setting (Figure 1A,B). The results of proteome analyses showed that 3619 proteins were identified and quantified. From CCO cells with or without SHVV cultured in the complete medium, 346 proteins were considered DEPs (Table S1), of which 135 were up-regulated and 211 were down-regulated (Figure 1C). Three hundred and seventy differentially expressed proteins (DEPs) were identified from the CCO cells with SHVV cultured in the glutamine-free medium or the complete medium (Table S2), containing 124 up-regulated proteins and 246 down proteins (Figure 1D). One hundred and forty-seven proteins were considered DEPs in both of the experimental groups (Figure 1E).

### 3.2. Subcellular Localization of Differentially Expressed Proteins (DEPs)

Proteins in vivo can play their roles only in specific subcellular positions, so predicting protein subcellular location is critical for understanding protein function and mechanism [44–46]. The DEPs were principally localized in cytoplasm, nuclei, and mitochondria in both experimental groups (Figures 2A and 3A).

Most negative-stranded RNA viruses replicate in the cytoplasm of the host cell, including the rabies virus [47], bunyavirus [48], Sendai virus [49], and others. Various cellular mechanisms involved in RNA viral replication, such as autophagy [50], also occur in the cytoplasm. The nucleus of the host cell also plays a critical role in RNA viral replication [51], for example, PTMs are associated with nuclear-cytoplasmic trafficking [52]. In addition, replication of several negative-stranded RNA viruses occurs in the nucleus of the host cell [53], such as influenza A virus [54]. Glutamine is associated with the TCA cycle, which occurs in the mitochondria of host cells and provides energy for RNA virus replication [55].

### 3.3. GO Analysis of Differentially Expressed Proteins (DEPs)

Gene ontology (GO) analyses of DEPs showed significantly enriched functions from biological processes (BP), cellular components (CC), and molecular functions (MF).

Biological process analysis revealed that the DEPs from CCO cells with or without SHVV cultured in the complete medium (Figure 2B) were mostly related to cellular processes, metabolic processes, single-organism processes, biological regulation, etc. Cellular component analysis indicated that the DEPs mainly participated in cells, cell parts, organelles and membranes. In the molecular functions category, binding contained the most DEPs, followed by catalytic activity. Furthermore, the DEPs were mainly enriched in the regulation of nucleic acid-templated transcription, regulation of RNA biosynthetic process, signal transduction, etc. (Figure 2C).

The analysis results of the DEPs between CCO cells cultured in the complete medium or the glutamine-free medium (Figure 3B) were similar to the above. DEPs were involved in cellular processes, metabolic processes, single-organism processes, biological regulation processes and regulation of biological processes in biological process analysis. In the cellular component analysis, DEPs were largely related to cells, cell parts, organelles, and membranes, and the major molecular functions were binding and catalytic activity. The DEPs were primarily enriched in ubiquitin ligase complex, cullin-RING ubiquitin ligase complex, transcription factor activity, transcription factor binding, transcription cofactor activity, etc. (Figure 3C).

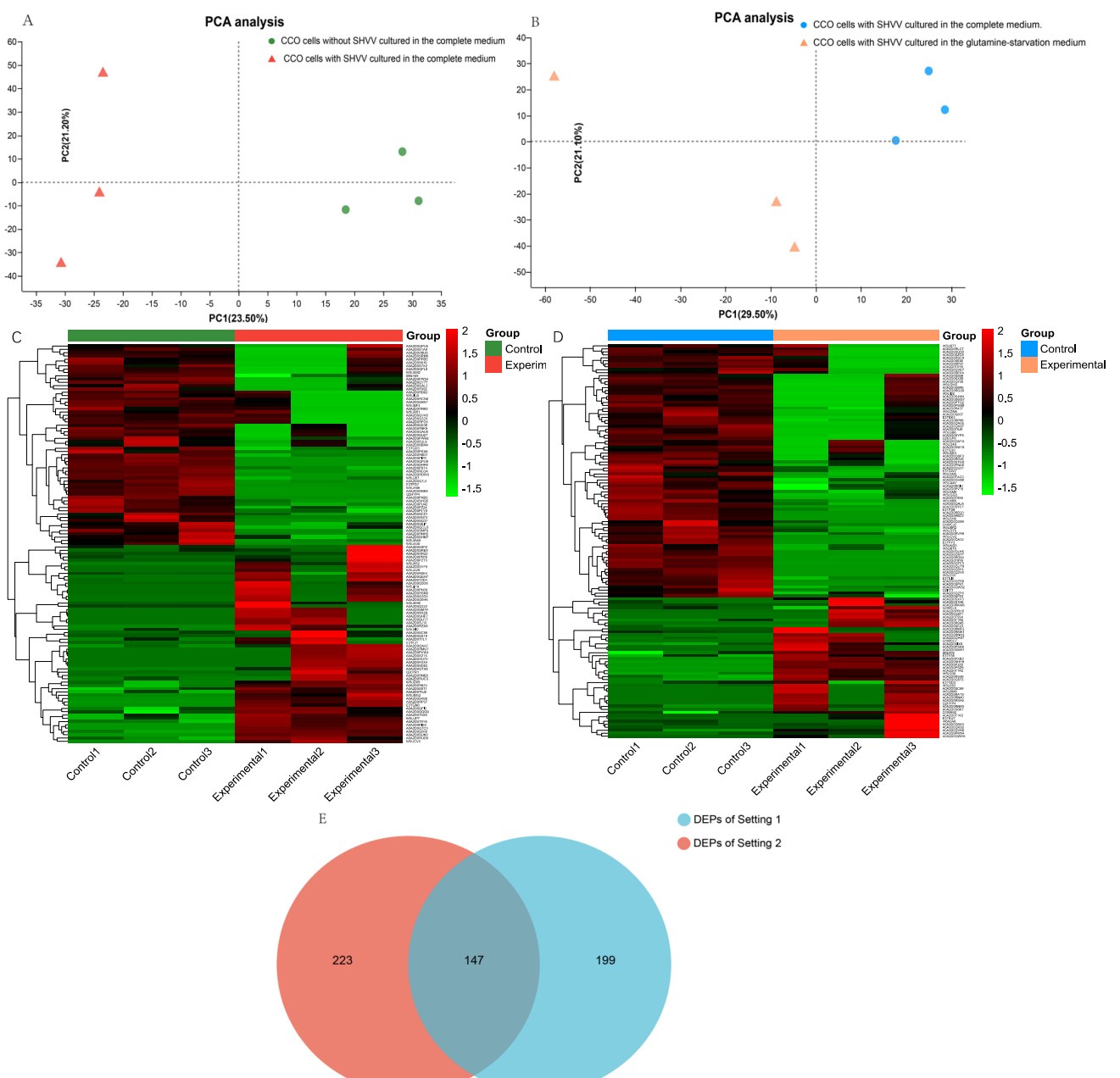

**Figure 1.** (**A**,**B**) The principal component analysis (PCA) models of each experimental group. (**A**) Samples from CCO cells with or without SHVV cultured in the complete medium. (**B**) Samples from CCO cells with SHVV cultured in the glutamine-starvation medium or the complete medium. (**C**,**D**) Heatmaps of DEPs identified with fold change >2 or <0.5, *p*-value < 0.05). (**C**) Heatmap of experimental setting 1. Control: DEPs from CCO cells without SHVV were cultured in the complete medium. Experimental: DEPs from CCO cells with SHVV were cultured in the complete medium. (**D**) Heatmap of experimental setting 2. Control: DEPs from CCO cells with SHVV cultured in the complete medium. Experimental: DEPs from CCO cells with SHVV cultured in the glutamine-free medium (**E**) Venn diagram of DEPs between the two experimental groups.

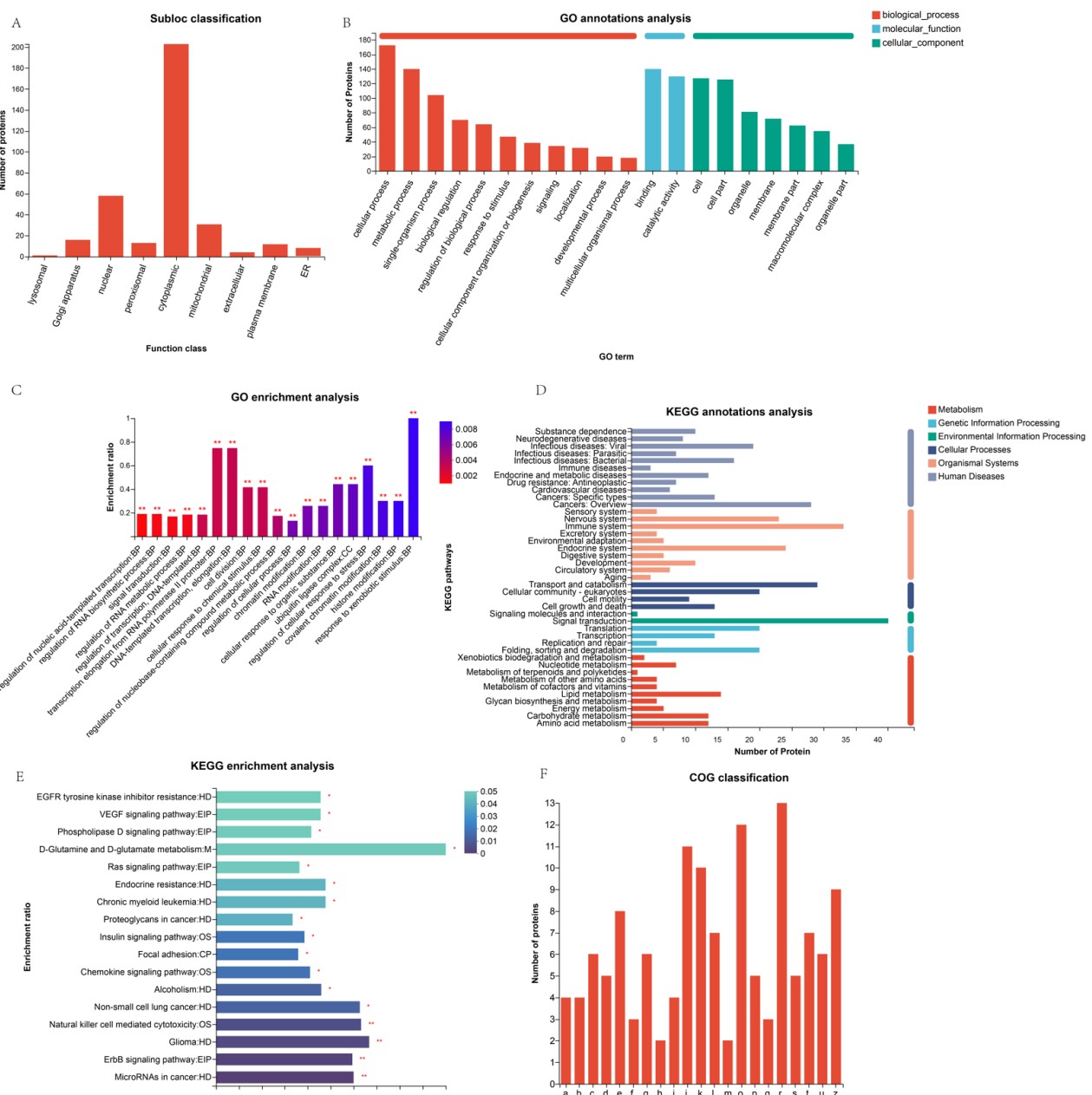

**Figure 2.** Bioinformatics analysis of DEPs from CCO cells with or without SHVV cultured in the complete medium. (**A**) Subcellular localization of DEPs. (**B**) GO annotations analysis of DEPs. (**C**) GO enrichment analysis of DEPs. (**D**) KEGG annotations analysis of DEPs. (**E**) KEGG enrichment analysis of DEPs. (**F**) COG analysis of DEPs. (a) RNA processing and modification; (b) Chromatin structure and dynamics; (c) Energy production and conversion; (d) Cell cycle control, cell division, chromosome partitioning; (e) Amino acid transport and metabolism; (f) Nucleotide transport and metabolism; (g) Carbohydrate transport and metabolism; (h) Coenzyme transport and metabolism; (i) Lipid transport and metabolism; (j) Translation, ribosomal structure and biogenesis; (k) Transcription; (l) Replication, recombination and repair; (m) Cell wall/membrane/envelope biogenesis; (o) Posttranslational modification, protein turnover, chaperones; (p) Inorganic ion transport and metabolism; (q) Secondary metabolites biosynthesis, transport and catabolism; (r) General function prediction only; (s) Function unknown; (t) Signal transduction mechanisms; (u) Intracellular trafficking, secretion, and vesicular transport; (z) Cytoskeleton. *, *p*-value < 0.05; **, *p*-value < 0.01.

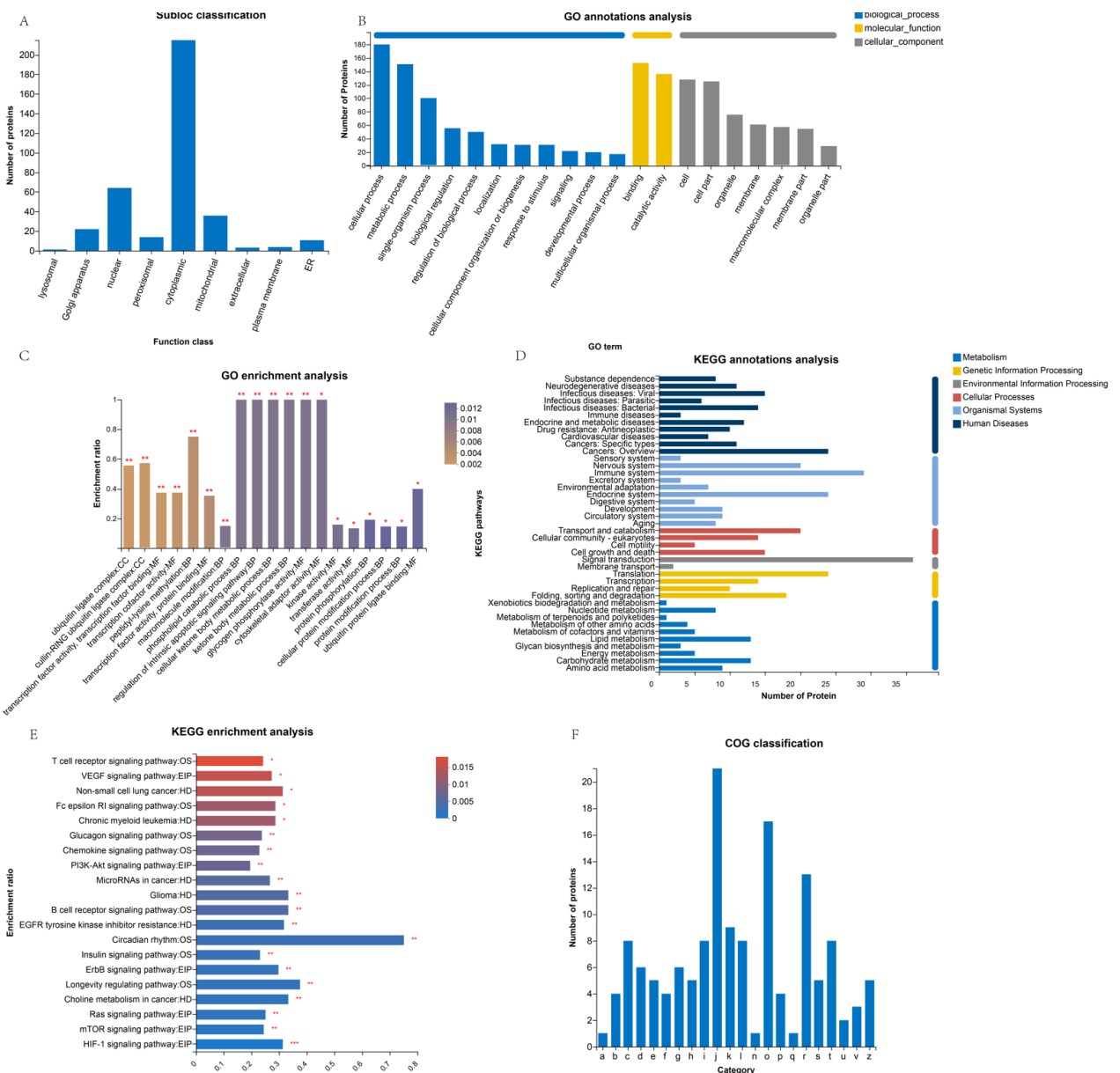

**Figure 3.** Bioinformatics analysis of DEPs from CCO cells with SHVV cultured in the glutamine-free medium or the complete medium. (**A**) Subcellular localization of DEPs. (**B**) GO annotations analysis of DEPs. (**C**) GO enrichment analysis of DEPs. (**D**) KEGG annotations analysis of DEPs. (**E**) KEGG enrichment analysis of DEPs. (**F**) COG analysis of DEPs. (a) RNA processing and modification; (b) Chromatin structure and dynamics; (c) Energy production and conversion; (d) Cell cycle control, cell division, chromosome partitioning; (e) Amino acid transport and metabolism; (f) Nucleotide transport and metabolism; (g) Carbohydrate transport and metabolism; (h) Coenzyme transport and metabolism;(i) Lipid transport and metabolism; (j) Translation, ribosomal structure and biogenesis; (k) Transcription; (l) Replication, recombination and repair; (n) Cell motility; (o) Posttranslational modification, protein turnover, chaperones; (p) Inorganic ion transport and metabolism; (q) Secondary metabolites biosynthesis, transport and catabolism; (r) General function prediction only; (s) Function unknown; (t) Signal transduction mechanisms; (u) Intracellular trafficking, secretion, and vesicular transport; (v) Defense mechanisms; (z) Cytoskeleton. (\*, *p*-value < 0.05; \*\*, *p*-value < 0.01; \*\*\*, *p*-value < 0.001).

GO analysis demonstrated that SHVV infection affected metabolism and signal transduction processes in CCO cells, as well as the binding of intracellular compounds and the activities of key enzymes, and these effects were attenuated by glutamine starvation.

### 3.4. KEGG Pathway Analysis of Differentially Expressed Proteins (DEPs)

KEGG pathway annotation and enrichment analyses were conducted to confirm the signaling pathways correlated with the DEPs and to make further investigation of the critical factors involved in how SHVV influences the CCO cells and glutamine starvation affects the replication of SHVV. All KEGG pathways were classified into six categories, which include metabolism, genetic information processing, environmental information processing, cellular processes, organismal systems, and human diseases.

The results of KEGG annotation analysis revealed that DEPs from CCO cells with or without SHVV cultured in the complete medium were primarily annotated to signal transduction, transport and catabolism, and immune systems (Figure 2D). The KEGG enrichment results showed that DEPs from CCO cells with or without SHVV cultured in the complete medium were enriched in 17 KEGG pathways (Figure 2E, Table 1), including pathways that related to RNA virus infection, such as the ErbB signaling pathway [56] and natural killer cell-mediated cytotoxicity [57]. Furthermore, the annotation analysis also revealed that DEPs participated in the RIG-I-like receptor signaling pathway [58,59], Toll-like receptor signaling pathway [59,60], NF-kappa B signaling pathway [61,62], PI3K-Akt signaling pathway [63,64], MAPK signaling pathway [65,66], and other pathways that are associated with the antiviral response after RNA virus infection and RNA viral pathogenicity (Table 2).

**Table 1.** KEGG enriched pathways associated with the pathogenicity of SHVV.

| KEGG Description | *p*-Value |
|---|---|
| MicroRNAs in cancer | 0.0011 |
| ErbB signaling pathway | 0.0022 |
| Glioma | 0.0042 |
| Natural killer cell mediated cytotoxicity | 0.0056 |
| Non-small cell lung cancer | 0.0120 |
| Alcoholism | 0.0123 |
| Chemokine signaling pathway | 0.0168 |
| Focal adhesion | 0.0176 |
| Insulin signaling pathway | 0.0183 |
| Proteoglycans in cancer | 0.0367 |
| Chronic myeloid leukemia | 0.0383 |
| Endocrine resistance | 0.0383 |
| Ras signaling pathway | 0.0448 |
| D-Glutamine and D-glutamate metabolism | 0.0449 |
| Phospholipase D signaling pathway | 0.0452 |
| VEGF signaling pathway | 0.0459 |
| EGFR tyrosine kinase inhibitor resistance | 0.0459 |

The top 20 relevant KEGG pathways enriched with DEPs between CCO cells cultured in the complete medium or the glutamine-free medium were shown in Figure 3E and Table 3, including KEGG pathways that are related to RNA virus replication, such as the Ras signaling pathway [67,68], PI3K-Akt signaling pathway [64,69–75], mTOR signaling pathway [56,73,76,77], and HIF-1 signaling pathway [56]. The KEGG annotation results revealed that the signal transduction pathway, transport and catabolism pathway, translation pathway, and immune systems pathway contained the most DEPs (Figure 3D). Some of the subpathways have been proved to participate in the replication of RNA viruses, including the MAPK signaling pathway [65,78], RIG-I-like receptor signaling pathway [79,80], NF-kappa B signaling pathway [81,82], etc. (Table 4).

**Table 2.** Genes involved in some KEGG annotated pathways associated with the pathogenicity of SHVV.

| Proteins | Gene | Protein Fold Change | Subpathway | KEGG Subpathway *p*-Value | Superpathway |
|---|---|---|---|---|---|
| A0A2D0PVW4 | ACTB_G1 ↑ | >32 | RIG-I-like receptor signaling pathway | 0.4424 | immune system |
| A0A2D0QBZ9 | TRAF2 ↓ | <0.00001 | | | |
| A0A2D0RKM8 | AKT ↑ | >32 | Toll-like receptor signaling pathway | 0.9202 | |
| A0A2D0R0K4 | SHC1 ↑ | 3.652 | | | |
| A0A2D0QNJ3 | PPM1A, PP2CA ↑ | 2.457 | | | |
| A0A2D0QFC3 | CASP3 ↑ | >32 | | | |
| A0A2D0RBZ2 | PPP3C, CAN ↑ | >32 | MAPK signaling pathway | 0.1960 | |
| A0A2D0RKM8 | AKT ↑ | >32 | | | |
| A0A2D0PU15 | GRB2 ↓ | <0.00001 | | | |
| A0A2D0QBZ9 | TRAF2 ↓ | <0.00001 | | | |
| A0A2D0SZV8 | CRK, CRKII ↓ | <0.00001 | | | |
| A0A2D0T445 | ERC1, CAST2, ELKS ↓ | <0.00001 | NF-kappa B signaling pathway | 0.1781 | Signal transduction |
| A0A2D0SJ10 | PLCG1 ↑ | >32 | | | |
| A0A2D0QBZ9 | TRAF2 ↓ | <0.00001 | | | |
| A0A2D0SN73 | ERBB2IP, ERBIN ↓ | <0.00001 | | | |
| W5UJ57 | ITGA5, CD49e ↓ | 0.4468 | | | |
| A0A2D0RKM8 | AKT ↑ | >32 | | | |
| A0A2D0PSX8 | PTK2, FAK ↓ | <0.00001 | PI3K-Akt signaling pathway | 0.3448 | |
| A0A2D0QAU5 | GNB2 ↓ | <0.00001 | | | |
| A0A2D0S1J2 | RAPTOR ↓ | <0.00001 | | | |
| E3TE89 | GNG5 ↓ | <0.00001 | | | |
| W5UHZ2 | PPP2R5 ↓ | <0.00001 | | | |

Note: "↑": up-regulated, "↓": down-regulated.

**Table 3.** KEGG enriched pathways associated with the replication of SHVV.

| KEGG Description | *p*-Value |
|---|---|
| HIF-1 signaling pathway | 0.0005 |
| mTOR signaling pathway | 0.0017 |
| Ras signaling pathway | 0.0022 |
| Choline metabolism in cancer | 0.0025 |
| Longevity regulating pathway | 0.0026 |
| ErbB signaling pathway | 0.0028 |
| Insulin signaling pathway | 0.0030 |
| Circadian rhythm | 0.0033 |
| EGFR tyrosine kinase inhibitor resistance | 0.0033 |
| B cell receptor signaling pathway | 0.0051 |
| Glioma | 0.0051 |
| MicroRNAs in cancer | 0.0058 |
| PI3K-Akt signaling pathway | 0.0072 |
| Chemokine signaling pathway | 0.0073 |
| Glucagon signaling pathway | 0.0081 |
| Chronic myeloid leukemia | 0.0116 |
| Fc epsilon RI signaling pathway | 0.0116 |
| Non-small cell lung cancer | 0.0141 |
| VEGF signaling pathway | 0.0147 |
| T cell receptor signaling pathway | 0.0171 |

*3.5. COG Analysis of Differentially Expressed Proteins (DEPs)*

The COG database was used for protein homologous classification. COG annotation was run to evaluate the effects of different protein expressions on biological function.

**Table 4.** Genes involved in some KEGG annotated pathways associated with the replication of SHVV.

| Proteins | Gene | Protein Fold Change | Subpathway | KEGG Subpathway *p*-Value | Superpathway |
|---|---|---|---|---|---|
| A0A2D0PSX8 | PTK2, FAK ↑ | >32 | | | |
| A0A2D0QAU5 | GNB2 ↑ | >32 | | | |
| E3TE89 | GNG5 ↑ | >32 | | | |
| A0A2D0Q294 | PPP2R5 ↓ | <0.00001 | | | |
| A0A2D0QC90 | PRKAA, AMPK ↓ | <0.00001 | | | |
| A0A2D0QGD6 | EIF4E ↓ | <0.00001 | PI3K-Akt signaling pathway | 0.0072 | |
| A0A2D0QH87 | RELA ↓ | <0.00001 | | | |
| A0A2D0QR45 | EIF4E ↓ | <0.00001 | | | |
| A0A2D0QVE7 | PDPK1 ↓ | <0.00001 | | | |
| A0A2D0RKM8 | AKT ↓ | <0.00001 | | | |
| A0A2D0RSR0 | GNB4 ↓ | <0.00001 | | | |
| W5U9B8 | MLST8, GBL ↓ | <0.00001 | | | |
| A0A2D0SZV8 | CRK, CRKII ↑ | >32 | | | |
| A0A2D0PJF0 | RASGRF2 ↓ | <0.00001 | | | |
| A0A2D0QH87 | RELA ↓ | <0.00001 | MAPK signaling pathway | 0.2262 | Signal transduction |
| A0A2D0R0K4 | SHC1 ↓ | <0.00001 | | | |
| A0A2D0RBZ2 | PPP3C, CAN ↓ | <0.00001 | | | |
| A0A2D0RKM8 | AKT ↓ | <0.00001 | | | |
| E3TEE7 | RBX1, ROC1 ↓ | <0.00001 | | | |
| E3TE71 | ATPeV1D, ATP6M ↑ | 2.076 | | | |
| A0A2D0PUT4 | MIOS, MIO ↓ | <0.00001 | | | |
| A0A2D0QC90 | PRKAA, AMPK ↓ | <0.00001 | | | |
| A0A2D0QGD6 | EIF4E ↓ | <0.00001 | | | |
| A0A2D0QL93 | TELO2, TEL2 ↓ | <0.00001 | mTOR signaling pathway | 0.0017 | |
| A0A2D0QR45 | EIF4E ↓ | <0.00001 | | | |
| A0A2D0QVE7 | PDPK1 ↓ | <0.00001 | | | |
| A0A2D0RKM8 | AKT ↓ | <0.00001 | | | |
| E3TF29 | RRAGC_D ↓ | <0.00001 | | | |
| W5U9B8 | MLST8, GBL ↓ | <0.00001 | | | |
| A0A2D0QH87 | RELA ↓ | <0.00001 | NF-kappa B signaling pathway | 0.4632 | |
| A0A2D0SJ10 | PLCG1 ↓ | <0.00001 | | | |
| A0A2D0Q3G1 | PIN1 ↓ | <0.00001 | | | |
| A0A2D0QH87 | RELA ↓ | <0.00001 | RIG-I-like receptor signaling pathway | 0.0599 | Immune system |
| A0A2D0R0L0 | OTUD5, DUBA ↓ | <0.00001 | | | |
| W5U5F5 | IRF3 ↓ | <0.00001 | | | |

Note: "↑": up-regulated, "↓": down-regulated.

From CCO cells with or without SHVV cultured in the complete medium, the DEPs were divided into 21 specific categories according to the COG analysis (Figure 2F). The main functional categories were translation, ribosomal structure and biogenesis, posttranslational modification, protein turnover, chaperones, and general function prediction only, followed by transcription, cytoskeleton, amino acid transport and metabolism, and replication, recombination and repair.

For DEPs between CCO cells cultured in the complete medium or the glutamine-free medium, translation, ribosomal structure and biogenesis occupied the maximum proportion, with posttranslational modification, protein turnover, chaperones, and general function prediction only next (Figure 3F).

The results suggested that viral infection caused adverse effects on cellular mechanisms in host cells, such as ribosome biosynthesis, translation, and post-translational modifications, which in turn were rescued via glutamine starvation. It has been shown that ribosomal proteins are vital to RNA virus replication [83], as the virus inhibits host cell protein translation [84] and exploits host cellular translation machinery for its efficient replication [85]. Some RNA viruses induce mRNA degradation in host cells [84]. All of these were coincident with COG analysis.

### 3.6. Protein–Protein Interaction Network of Differentially Expressed Proteins (DEPs)

To explain the interactions of the DEPs in both experimental groups, the DEPs in each set were analyzed based on the STRING database (Supplementary Tables S1 and S2). The proteins with higher degrees were considered the key proteins [86]. The PPI network of experimental group 1 (CCO cells with or without SHVV cultured in the complete medium) contained 261 proteins and the PPI network of experimental group 2 (CCO cells with SHVV cultured in the glutamine-free medium or the complete medium) contained 283 proteins (Figure 4). The top 100 proteins selected by the combined score in the two groups were listed in Tables S3 and S4. The comparison of the two PPI networks revealed that several key proteins expressed different regulations (Table 5) between experimental group 1 (CCO cells with or without SHVV cultured in the complete medium) and experimental group 2 (CCO cells with SHVV cultured in the glutamine-free medium or the complete medium).

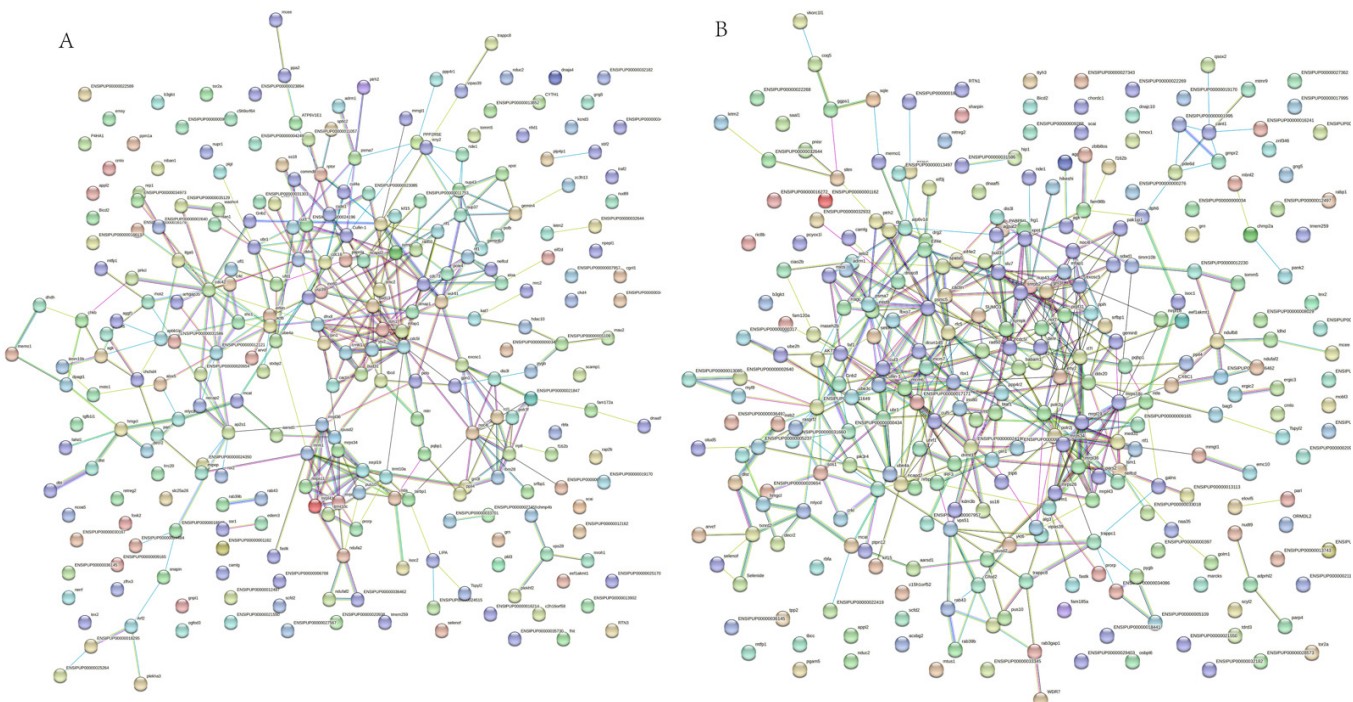

**Figure 4.** (**A**) PPI network of differentially expressed proteins (DEPs) from CCO cells with or without SHVV cultured in the complete medium. (**B**) PPI network of differentially expressed proteins (DEPs) from CCO cells with SHVV cultured in the glutamine-starvation medium or the complete medium. The highlighted nodes were involved in both PPI networks.

**Table 5.** Prediction of proteins associated with SHVV replication.

| UniProtKB | Protein Name | Gene | Regulation in Group 1 | Protein | | Regulation in Group 2 | Protein | |
|---|---|---|---|---|---|---|---|---|
| | | | | Fold Change | *p*-Value | | Fold Change | *p*-Value |
| A0A2D0RR20 | cell division cycle 5-like protein isoform X1 | CDC5L | Up | >32 | <0.00001 | Down | <0.00001 | <0.00001 |
| A0A2D0PN16 | guanine nucleotide-binding protein-like 3-like protein | GNL3L | Up | >32 | <0.00001 | Down | <0.00001 | <0.00001 |
| A0A2D0RJY2 | Hydroxymethylglutaryl-CoA lyase | HMGCL | Up | >32 | <0.00001 | Down | <0.00001 | <0.00001 |
| A0A2D0PW54 | nucleoporin Nup43 | NUP43 | Down | <0.00001 | <0.00001 | Up | >32 | <0.00001 |
| W5U919 | Proteasome subunit alpha type | PSMA7 | Down | <0.00001 | <0.00001 | Up | >32 | <0.00001 |
| A0A2D0RMF5 | intersectin-2-like isoform X1 | ITSN | Down | <0.00001 | <0.00001 | Up | >32 | <0.00001 |
| A0A2D0PSX8 | Non-specific protein-tyrosine kinase | PTK2 | Down | <0.00001 | <0.00001 | Up | >32 | <0.00001 |
| W5U6X2 | 28S ribosomal protein S34, mitochondrial | MRPS34 | Down | <0.00001 | <0.00001 | Up | >32 | <0.00001 |
| A0A2D0SA67 | 39S ribosomal protein L43, mitochondrial | MRPL43 | Down | <0.00001 | <0.00001 | Up | >32 | <0.00001 |
| A0A2D0T9T5 | cullin-1 | CUL1 | Down | <0.00001 | <0.00001 | Up | >32 | <0.00001 |

The proteins were involved in pathways and mechanisms associated with the life cycle of RNA viruses. The protein encoded by ITSN is associated with endocytosis and intracellular signal transduction, the protein encoded by NUP43 is a nuclear pore complex associated protein, and the protein encoded by CDC5L is required for mRNA processing. The RNA virus processes its replication from entry into the cell to protein expression, which includes RNA virus being released to the host cytoplasm and translated to the host nucleus, viral mRNA synthesis and exportation, and translation. Cytoplasmic membrane-associated proteins contribute to receptor-mediated endocytosis, and the RNA virus is released into the cytoplasm via endocytosis [87–89] and transported into the nucleus via host nuclear pore complexes [90–92] (NPCs). Nuclear–cytoplasmic trafficking is important to RNA viral replication, and the nucleus is the site where the nuclear-replicating RNA virus genome replication and transcription occurs [93].

The protein encoded by PTK2 is related to ATP binding, and MRPS34 and MRPL43 encode mitochondrial ribosomal proteins that contribute to protein synthesis in the mitochondrion. Mitochondria are the main sites of ATP synthesis [94] and are also involved in cellular immunity [95]. ATP is the energy source for viral replication, and some RNA viruses evade host cell immunity and promote viral replication by interacting with host mitochondrial proteins [96].

Proteins encoded by CUL1 and PSMA7 are involved in protein ubiquitination. Ubiquitination, as one of the post-translational modifications, has been proven to be closely associated with RNA virus replication [97]. Ubiquitination is an important post-translational modification mechanism closely related to immune regulation [98], and the ubiquitin-proteasome system is a major protein degradation pathway. Studies have shown that RNA viruses can use the ubiquitin system to evade host immunity [99,100] and enhance viral replication [101–104] during viral infection. Cullin is critical to the activity of the E3 ligase complex [105–108], and cullin-1 encoded by CNL1 plays an important regulatory role in the SCF (SKP1-Cul1-F-box) E3 ubiquitin ligase complex [109] and contributes to various cellular processes [110,111]. It has been proven that RNA viral proteins utilize host cell ubiquitination mechanisms through the SKP1-CUL1-F-box E3 ligase complex [112].

Those proteins have been proven to participate in cellular mechanisms correlated to RNA viral replication and were therefore predicted to participate in SHVV pathogenesis via their effects on the replication of SHVV.

## 4. Discussion

We created a glutamine starvation condition for CCO cells with SHVV by using a glutamine-free medium, and CCO cells with SHVV cultured in a complete medium (with glutamine) were used as control. We have verified the regulation of glutamine starvation on SHVV replication in combination with previous studies. The regulatory roles of host cell proteins for viral replication are diverse. Hsp90 was up-regulated with SHVV infection (fold change = 32, $p$-value < 0.05) and significantly down-regulated in the condition of glutamine starvation (fold change = 0.00001, $p$-value < 0.05). A previous study demonstrated that Hsp90 is essential for SHVV replication, interacting with the SHVV protein to enhance its replication via stabilizing viral proteins [113]. AMPK was also significantly down-regulated in the condition of glutamine starvation and was considered to be a DEP associated with SHVV replication. AMPK is a hub for regulating host cell metabolism [114] and plays an important role in viral infection [115] and replication [116]. AMPK has been shown to be up-regulated after SHVV infection and has been proven to be a positive regulator of SHVV replication [28]. Down-regulation of AMPK results in up-regulation of IFN-$\alpha$, which inhibited SHVV replication via mediating the innate immune response [28]. Furthermore, DEPs were also involved in various signaling pathways and cellular mechanisms that regulate viral replication, such as autophagy and PTMs.

We also proposed the proteins related to SHVV replication, and the signaling pathways and cellular mechanisms they were involved in. AKT, a serine/threonine kinase, was significantly up-regulated in the CCO cells during SHVV infection and was notably

down-regulated in the CCO cells with SHVV cultured in the glutamine-free medium, compared with those cultured in the complete medium. AKT is a critical protein that is involved in multiple signaling pathways including the Toll-like receptor signaling pathway, PI3T-AKT signaling pathway, MAPK signaling pathway, and mTOR signaling pathway, and is a central regulator that acts on various cellular processes including cell survival [117], metabolism, growth, proliferation, and migration [118]. AKT is essential to the effective replication of RNA viruses [70,119,120] and is a vital regulator of the PI3K-Akt signaling pathway, which is activated during RNA viral infection [121] and is negatively related to autophagy [13,122–124], and autophagy inhibits the replication of SHVV in host cells [15,125]. In addition, AKT promotes viral replication via anti-apoptosis of the host cell [75,126] and plays a key role in activating the viral polymerase for replication of viral RNA synthesis [64,127]. Nevertheless, the role of AKT in SHVV replication has not been proven. Based on this study, we have formulated a hypothesis that AKT regulates SHVV replication (Figure 5), and that the L protein of SHVV is an RNA-dependent RNA polymerase.

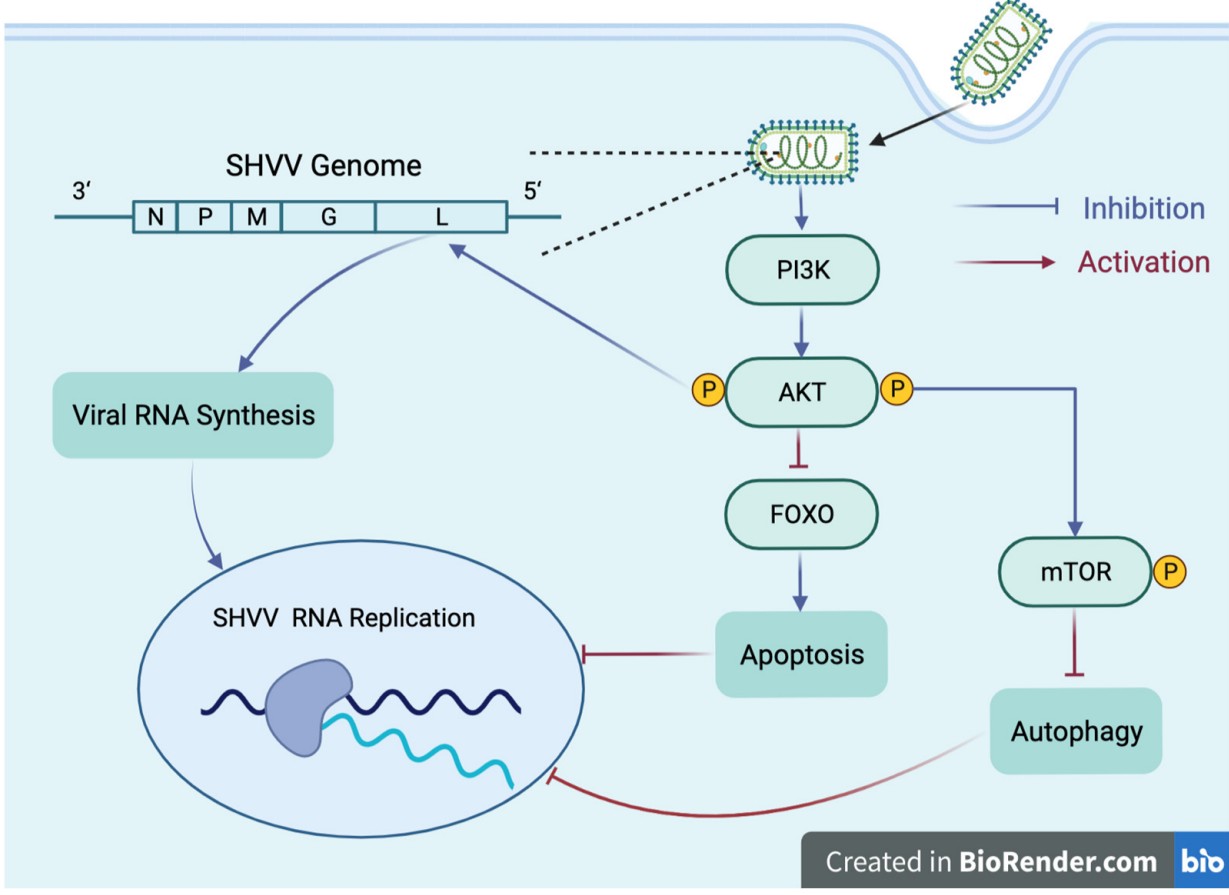

**Figure 5.** A hypothesis for the involvement of AKT in SHVV replication. AKT is activated during virus infection and promotes virus replication via anti-apoptosis and inhibiting autophagy. AKT also participates in RNA synthesis of SHVV via activating the viral RNA-dependent RNA polymerase. The illustration was generated using BioRender (https://biorender.com/) (accessed on 26 June 2022).

RNA viral replication needs a complicated regulatory network of protein interactions in host cells [3]. Overall, we explained SHVV pathogenicity and the effect of glutamine starvation on SHVV replication from regulatory factors, signaling pathways, and cellular mechanisms via quantitative proteomics analysis, and proposed the key proteins and signaling pathways related to SHVV replication. This could be beneficial to the future study of the SHVV replication mechanism and to the prevention of SHVV infection.

**Supplementary Materials:** The following supporting information can be downloaded at: https://www.mdpi.com/article/10.3390/fishes7060315/s1. Table S1: Setting1 DEPs, Table S2: Setting2 DEPs, Table S3: Setting1 Top 100 Nodes, Table S4: Setting2 Top 100 Nodes.

**Author Contributions:** Experimental design, M.T., L.S. and K.C.; data analysis, Y.L., writing—review and editing, J.L., revision, Y.Z., H.Z. and X.L. All authors have read and agreed to the published version of the manuscript.

**Funding:** This work was financially supported by the grants from Natural Science Foundation of Jiangsu Province (BK20210747) of Lindan Sun, and Guangdong South China Sea Key Laboratory of Aquaculture for Aquatic Economic Animals, Guangdong Ocean University (KFKT2019YB09) of Lindan Sun.

**Institutional Review Board Statement:** Not applicable. This study do not involve humans or animals.

**Data Availability Statement:** The data presented in this study are available on request from the corresponding author.

**Conflicts of Interest:** The authors declare no conflict of interest.

## Abbreviations

| | |
|---|---|
| RIPA Lysis Buffer | Radio Immunoprecipitation Assay Lysis buffer |
| Tris-HCl | TRIS hydrochloride |
| EDTA | Ethylene Diamine Tetraacetic Acid |
| NP-40 | Nonidet P 40 |
| SDS | Sodium dodecyl sulfate |
| PMSF | Phenylmethylsulfonyl fluoride |
| DTT | Dithiothreitol |

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
