# Peer review of "Effects of Glutamine Starvation on SHVV Replication by Quantitative Proteomics Analysis"

_fishes, doi:10.3390/fishes7060315_

Round 1

Reviewer 1 Report

Please refer to species (scientific) name for “snakehead fish” in the beginning of the text or in the Abstract.

Could you please comment/refer/indicate further information why the channel catfish ovary (CCO) cell line was specifically selected instead of a snakehead-fish cell line e.g., SSN-1 as the in vitro model approach for this work? Was it due to its advantage of being easily infected by SHVV, as mentioned on line 51?

Could you please clarify/inform the reader about the rationale behind this choice, as it has been demonstrated that glutamine deficiency in CCO cells resulted in restricted growth, with cell viability being much lower in glutamine-free medium (with supplemented glucose), as compared with glucose-free medium (supplemented with glutamine) (Slivac et al, 2008 in Cytotechnology, doi: 10.1007/s10616-008-9171-y). Indeed, previous studies on glutamine starvation conducted by the authors were conducted on SSN-1 fish cells. Why now this move into CCO fish cell lines?

At this point it could be tricky to induce glutamine starvation in this cell line to address the effects of glutamine starvation on the SHVV replication, despite using a control of the virus replicating in cells growing in complete medium. Without glutamine, the replication of the virus might be inhibited, but additionally the cells have also their growth inhibited due to glutamine starvation itself, contributing somehow for a “noisy background” /”baseline stress starvation proteome” . Could you comment on this, please, by giving your insight?

Line 15: “....(DEPs) Label-free.” Please revise, as “label-free” refers to the method of estimating relative protein abundance.

Line 19-20. Proteins “were” instead of “are”… significantly differentially expressed. Please revise the grammar (and terms) in the remaining text, in order to avoid words like “compositions” (line 32), when referring to biomolecules inside the cells.

Lines 61-62. Please complete by mentioning the label-free quantitative method used for estimating protein abundance e.g., emPAI, NSAF, spectral counting, TOP3 ion precursor intensity (Which one was used?). Please indicate it also on the Material and Methods section.

Lines 90-91. Please review the whole paragraph to become properly described, e.g., “… solubilized in 10 mM Tris(2-carboxyethyl)phosphine (TCEP)….”

No protease inhibitors were added?

Please change also “ after reduction, used….”. Use instead a description similar to “ protein samples were reduced with XX mM DTT, followed by alkylation with 55 mM iodoacetamide….”

I suggest a careful revision of the whole section of LC-MS/MS to adopt a high standard description of MS-based proteomics.

For instance, the database search description is not clear enough; please indicate the taxonomy ID used for search, the total number of entries present in the database; if it was used a NCBI and UniProt merged database; or if it was only used an UniProt database, since the list of proteins indicated on the tables correspond to UniProt accession numbers.

Lines 132-135, regarding statistical analyses, this section seems to be related with the statistical tests used for enrichment analyses, rather indicating the statistical tests used for testing the presence of differentially abundant proteins between controls and the experimental groups (e.g., a T-test, with FDR correction). This is only slightly mentioned on lines 112-114, and it should be indicated somewhere in the Material and Methods,  the type of statistical test used (parametric or no parametric), followed by the used adjusted p-value, for determining DEPs.

Line 116. Proteins “were” annotated instead of “are”.

Lines 128-129. Please indicate for what BIOGRID and REACTOME were used as references; BIOGRID was used for PPI network analysis (please detail the objective) inside Cytoscape? and REACTOME was used for signaling pathway analysis also inside Cytoscape? The whole paragraph (lines 126-131) is not clear.

Line 136. “..PPI network was “constructed” (?) instead of “contracted” by Cytoscape.

A final comment to the number of experimental replicates and number of LC-MS/MS runs (one run per sample? Or 2 runs per sample?) used in this work. It should be clearly stated in this section of Material and Methods, despite being indicated by the authors in the results (line 142).

Lines 143-149. This paragraph should be reformulated since the PCA analysis is not shown, and we cannot see how the distinct groups (and replicates) get separated/become differentiated. In this case, the 4 groups could have been taken together for analysis. Afterwards, this PCA information could be complemented/compared with the results obtained by the hierarchical clustering analysis, as shown in the heat map (Figure 1). This Figure is not very informative as we only see that we have up and down regulated in both groups, which can be represented by a simple Venn diagram. Here, the interesting thing would have been to see the proteins annotated with similar functions grouping together, but unfortunately we cannot see any protein name/annotation on the figure, or any comment on the text about it. 

Lines 201-211. Tables 1 and 2 should indicate the fold-enrichment value and the statistical test, followed by the FDR (p-adjusted) value. The titles of Tables 1 and 2 should be “KEGG enriched pathways….”

Line 243. It would be important to have a supplementary material with the information of these TOP100 proteins and the criteria (combined score based on ? ….) used in the PPI network analysis on STRING. Information should be provided about the significant fold-changes, their statistical test and p-values, and corresponding annotations according to the different databases used. This information should be really included as supplemental data of the main text. Otherwise, we may only guess it.

Lines 242-250 and Table 3. It is hard to follow these results without visualizing the networks, as it is not clear how both networks were compared. Based on the table, do you mean that the same protein has different regulation when comparing the 2 experimental settings? And the Column with the “description” corresponds to the annotation of the same protein? If so, in which database? I am not sure if I could follow correctly Table 3… I suggest modifying this table in order to become clear.

The title is not also clear when related with the “prediction” of proteins associated with SHVV, as it seems that some model was used for prediction, and this was not the case. The same is true (lines 281-283, 301) about “prediction”. These are the proteins you hypothesize to participate in SHVV pathogenesis, so it is not entirely correct to say you “predict” because predictions are usually based on mathematical models e.g., machine learning.

As a final comment, proteomics data should be deposited on a public repository (e.g., PRIDE), through ProteomeXChange, thus following the MIAPE recommendations for the publication of Proteomics articles.

Author Response

Dear Editors,

Thank you for taking time out of your busy schedule to review our manuscript. We have studied the valuable comments carefully and tried our best to revise the manuscript. The point-to-point response to the reviewers’ comments are listed as follows:

Reviewer 1

Comment 1: Please refer to species (scientific) name for “snakehead fish” in the beginning of the text or in the Abstract.

Response 1: Thanking the reviewer for pointing this out. We have added scientific name of “snakehead fish” in line 11.

Comment 2: Could you please comment/refer/indicate further information why the channel catfish ovary (CCO) cell line was specifically selected instead of a snakehead-fish cell line e.g., SSN-1 as the in vitro model approach for this work? Was it due to its advantage of being easily infected by SHVV, as mentioned on line 51?

Response 2: Yes, CCO cells are very sensitive to SHVV infection according to reference 27 in the manuscript. But there are no studies related to the proteomics analysis of CCO infected by SHVV. This research could expand more data on the relationship between the virus and its host.

Feng, S.; Su, J.; Lin, L.; Tu, J. Development of a reverse genetics system for snakehead vesiculovirus (SHVV). Virology 2019, 526, 32-37

Comment 3: Could you please clarify/inform the reader about the rationale behind this choice, as it has been demonstrated that glutamine deficiency in CCO cells resulted in restricted growth, with cell viability being much lower in glutamine-free medium (with supplemented glucose), as compared with glucose-free medium (supplemented with glutamine) (Slivac et al, 2008 in Cytotechnology, doi: 10.1007/s10616-008-9171-y). Indeed, previous studies on glutamine starvation conducted by the authors were conducted on SSN-1 fish cells. Why now this move into CCO fish cell lines?

Response 3: Thank you for the information and literature that CCO growth require glutamine. Our study also found that SHVV reproduction require glutamine. For the purpose of exploring the carbon source, nitrogen source and energy allocation of glutamine for SHVV propagation and CCO cell survival,this program study the relationship of glutamine deficiency、SHVV infection and pathways of CCO cells further.

Slivac showed CCO utilize glutamine to enter the TCA cycle producing energy and synthesize proteins to promote its own growth. SHVV also use glutamine to enter the TCA cycle for proliferation (Sun et al, 2016). We wanted to explore viruses and hosts how to allocate resources at different concentrations of glutamine? AS so we did CCO cells proteomics infected by SHVV in the two extreme conditions (glutamine sufficient and glutamine deficient) to look for the difference in proteins and pathways, in order to further analyze the glutamine consumption pathway of host and virus. This article is a preliminary exploration, and we will do more research later.

Sun Lindan, et al. Glutamine is required for snakehead fish vesiculovirus propagation via replenishing the tricarboxylic acid cycle. The Journal of General Virology: A Federation of European Miorobiological Societies Journal 2016, 2849-2855.

Comment 4: At this point it could be tricky to induce glutamine starvation in this cell line to address the effects of glutamine starvation on the SHVV replication, despite using a control of the virus replicating in cells growing in complete medium. Without glutamine, the replication of the virus might be inhibited, but additionally the cells have also their growth inhibited due to glutamine starvation itself, contributing somehow for a “noisy background” /”baseline stress starvation proteome”. Could you comment on this, please, by giving your insight?

Response 4: Thank you for this serious question. Glutamine deficiency results in decreasing cell viability and reducing SHVV replication. In this paper, the proteomic analysis is initially looking for pathways affected by both cells and viruses. Comparing the two proteomic analysis of virus infection and glutamine supplementation, we identified the possible CCO differential proteins of glutamine affecting viral replication. Next study we will seriously consider your proposed the " noisy background " problem, [13C5/15N2] glutamine marked N and C by isotope tracer technology will be used to study the flow of the N and C, to further study the problem of " baseline stress starvation proteome ", solve the ultimate goal of research: SHVV and glutamine in the CCO allocation problem.

Comment 5: Line 15: “.... (DEPs) Label-free.” Please revise, as “label-free” refers to the method of estimating relative protein abundance.

Response 5: Thanking the reviewer for pointing this out, we have deleted “label-free” in line 15 for clarity and precision.

Comment 6: Line 19-20. Proteins “were” instead of “are”… significantly differentially expressed. Please revise the grammar (and terms) in the remaining text, in order to avoid words like “compositions” (line 32), when referring to biomolecules inside the cells.

Response 6: Thanks for your suggestions. We have carefully revised the grammar and term problems in the article. Here is what we have revised:

Line 19: are—were

Line 32: compositions—biomolecules

Line 121: are—were

Line 240: is—was

Comment 8: Lines 90-91. Please review the whole paragraph to become properly described, e.g., “… solubilized in 10 mM Tris(2-carboxyethyl) phosphine (TCEP)….”

Comment 10: Please change also “after reduction, used….”. Use instead a description similar to “protein samples were reduced with XX mM DTT, followed by alkylation with 55 mM iodoacetamide….”

Response 8 & 10: Thanks for the suggestion of the reviewer, we have changed the description of Protein alkylation and trypsin digestion on lines 92-97 to make the description more accurate and clearer.

Comment 9: No protease inhibitors were added?

Response 9: Thanking the reviewer for pointing this out. We have filled in more details for RIPA Lysis Buffer in line 85, to ensure the reproducibility of the experiment.

Comment 7: Lines 61-62. Please complete by mentioning the label-free quantitative method used for estimating protein abundance e.g., emPAI, NSAF, spectral counting, TOP3 ion precursor intensity (Which one was used?). Please indicate it also on the Material and Methods section.

Comment 11: I suggest a careful revision of the whole section of LC-MS/MS to adopt a high standard description of MS-based proteomics.

Response 7 & 11: According to your suggestion, we have added more details for protein quantification, and redescribed the LC-MS /MS section (lines 99-111) for expressing more clearly and accurately.

Comment 12: For instance, the database search description is not clear enough; please indicate the taxonomy ID used for search, the total number of entries present in the database; if it was used a NCBI and UniProt merged database; or if it was only used an UniProt database, since the list of proteins indicated on the tables correspond to UniProt accession numbers.

Response 12: Thank you for pointing this out. We used Uniprot database for database search finally, and we have modified the description in Materials and Methods (Line 114-117). The taxonomy ID and the total number of entries in the database have been added.

Comment 13: Lines 132-135, regarding statistical analyses, this section seems to be related with the statistical tests used for enrichment analyses, rather indicating the statistical tests used for testing the presence of differentially abundant proteins between controls and the experimental groups (e.g., a T-test, with FDR correction). This is only slightly mentioned on lines 112-114, and it should be indicated somewhere in the Material and Methods, the type of statistical test used (parametric or no parametric), followed by the used adjusted p-value, for determining DEPs.

Response 13: Thanking for the reviewer’s careful correction. We have corrected the statement about the statistical test for enrichment analysis, while we have added the statistical test for determining differentially expressed proteins, and corrected the expression of p-values in lines 118 and 158.

Comment 14: Line 116. Proteins “were” annotated instead of “are”.

Response 14: Thanking for kindly correction. We have fixed this grammar error and detailed instruction is included in Response 6.

Comment 15: Lines 128-129. Please indicate for what BIOGRID and REACTOME were used as references; BIOGRID was used for PPI network analysis (please detail the objective) inside Cytoscape? and REACTOME was used for signaling pathway analysis also inside Cytoscape? The whole paragraph (lines 126-131) is not clear.

Response 15: Thank you for pointing out this problem. We mainly used the protein interaction information obtained from STRING database to construct PPI network, and also referred to the protein interactions information in BIOGRID database. We also performed a simple functional enrichment analysis of proteins including the REACTOME pathways in the STRING database, but this is not the focus of the results described, so we have removed this point from the manuscript. We have revised the description of the PPI network construction (lines 134-136) to make it more accurate and match our figures better.

Comment 16: Line 136. “..PPI network was “constructed” (?) instead of “contracted” by Cytoscape.

Response 16: Thank you for pointing out this mistake. We have corrected the spelling error.

Comment 17: A final comment to the number of experimental replicates and number of LC-MS/MS runs (one run per sample? Or 2 runs per sample?) used in this work. It should be clearly stated in this section of Material and Methods, despite being indicated by the authors in the results (line 142).

 Response 17: Thank you for your suggestion. We have added details about the experimental settings in line 82 and LC-MS/MS assay in line 101.

Comment 18: Lines 143-149. This paragraph should be reformulated since the PCA analysis is not shown, and we cannot see how the distinct groups (and replicates) get separated/become differentiated. In this case, the 4 groups could have been taken together for analysis. Afterwards, this PCA information could be complemented/compared with the results obtained by the hierarchical clustering analysis, as shown in the heat map (Figure 1). This Figure is not very informative as we only see that we have up and down regulated in both groups, which can be represented by a simple Venn diagram. Here, the interesting thing would have been to see the proteins annotated with similar functions grouping together, but unfortunately, we cannot see any protein name/annotation on the figure, or any comment on the text about it. 

Response 18: Thank you for pointing it out. We have made improvements and supplements to the sample analysis section. We added PCA figures based on all proteins in samples and DEPs in each experimental settings (Figure 1A Figure 1B). At the same time, we added a Venn diagram based on DEPs of the two experimental groups and annotated proteins in the heatmaps (Figure 1E).

Comment 19: Lines 201-211. Tables 1 and 2 should indicate the fold-enrichment value and the statistical test, followed by the FDR (p-adjusted) value. The titles of Tables 1 and 2 should be “KEGG enriched pathways….”

Response 19: Thank you for your suggestion. We showed key genes participated in some of the KEGG annotation pathways in Tables 1 and Table 2. We empirically screened out the KEGG annotation pathways because they are related to RNA viral replication although the P-values of some pathways were >0.05, and the genes involved in them are also the focus of our possible subsequent studies. According to your suggestion, we added P-values of KEGG pathways (Table 1 & Table 3) and added KEGG pathways with P-value<0.05 to Tables 2 and 4 (formerly Table 1 & Table 2), revised the table headers. Meanwhile, we added KEGG and GO enrichment figures (Figure 2 & Figure 3), and redescribed the part of KEGG and GO analysis to make our expressions more precise and highlighted.

Comment 20: Line 243. It would be important to have a supplementary material with the information of these TOP100 proteins and the criteria (combined score based on? ….) used in the PPI network analysis on STRING. Information should be provided about the significant fold-changes, their statistical test and p-values, and corresponding annotations according to the different databases used. This information should be really included as supplemental data of the main text. Otherwise, we may only guess it.

Response 20: Thank you for your suggestion. We have added PPI network diagram (Figure 4) in the main manuscript and submitted the table with relevant information as a supplementary. At the same time, we added information of all DEPs in the supplementary table to provide more reliable data and conclusions.

Comment 21: Lines 242-250 and Table 3. It is hard to follow these results without visualizing the networks, as it is not clear how both networks were compared. Based on the table, do you mean that the same protein has different regulation when comparing the 2 experimental settings? And the Column with the “description” corresponds to the annotation of the same protein? If so, in which database? I am not sure if I could follow correctly Table 3… I suggest modifying this table in order to become clear.

Response 21: Thank you for your question and suggestion. Your understanding is correct, the proteins in Table 5 (formerly Table 3) were screened for proteins that had opposite regulatory effects in the two experimental settings because group 1 was set to study viral pathogenicity and group 2 was set to study the effects that glutamine inhibits viral replication. We think that a protein that is up/down regulated because of viral infection is likely to be closely related to viral infection or viral replication if it is down/up regulated in the experimental group that glutamine was added. To make the presentation clearer, we have taken your suggestion and added the Fold-Change and P-value of the protein in each group in Table 5 (formerly Table 3), and we have focused on the description of the protein function in the main manuscript.

Comment 22: The title is not also clear when related with the “prediction” of proteins associated with SHVV, as it seems that some model was used for prediction, and this was not the case. The same is true (lines 281-283, 301) about “prediction”. These are the proteins you hypothesize to participate in SHVV pathogenesis, so it is not entirely correct to say you “predict” because predictions are usually based on mathematical models e.g., machine learning.

Response 22: Thanking for the reviewer’s careful correction. In this section (Line 319-335), we formulated a hypothesis for the regulation of SHVV replication based on the key signaling pathways and key proteins that were screened in our study. The validation and study of mechanisms would be the focus of our subsequent research. However, we used an unprecise word in this section, and we have revised the word "prediction" to "hypothesis" and "speculation" according to your suggestion to make our description precise and proper (line 366 and line 381). Thank you very much for your help in weighing words in our manuscript.

Comment 23: As a final comment, proteomics data should be deposited on a public repository (e.g., PRIDE), through ProteomeXChange, thus following the MIAPE recommendations for the publication of Proteomics articles.

Response 23: Thank you for your reminder. We will submit the data as soon as the manuscript is confirmed to be published.

Furthermore, we adjusted the overall color of some figures to make it more uniform and beautiful..

We have carefully checked our manuscript and revised it according to your suggestion. We hope that our responses and edits are satisfactory. Please contact us if there are any questions. We are looking forward to hearing from you soon.

Thanks again for your hard work!

Best wishes!

Reviewer 2 Report

Dear Editor,

The authors of the manuscript entitled “Effects of the glutamine starvation on SHVV replication by quantitative proteomics analysis” (Junlin Liu) presents a proteomic analysis of Channel Catfish Ovary (CCO) cells infected with SHVV cultivated in glutamine-free or complete medium to assess how glutamine starvation inhibits the SHVV replication.

Τhe manuscripts’ objects are interesting, the authors have collected and analyze the proteomics data adequately. Therefore, the manuscript could be accepted for publication after minor revisions. My detailed comments for the authors to consider are provided below:

1.      Page 2, lines 67-73: The authors should provide more detailed protocols for virus isolation and cell cultures or add the appropriate references.

2.      Page 2, lines 84-89: Please define acronyms.

3.      Page 2, section 2.4: Please rephrase, this paragraph is very confusing. Is ug used instead of ng?

4.      Page 3, lines 126-127: the PPI network diagram should be included in the results section.

5.      Page 9, line 289: Hsp90 is not included in any of the presented tables. Why and how it is commented here?

6.      Page 10, lines 315-317: The proposed mechanism seems interesting, however the authors should provide a more detailed discussion on it, i.e. explain the rationale behind figure 4, possible alternative scenario, etc.

Author Response

Dear Editors,

Thank you for taking time out of your busy schedule to review our manuscript. We have studied the valuable comments carefully and tried our best to revise the manuscript. The point-to-point response to the reviewers’ comments are listed as follows:

Reviewer 2

Comment 1: Page 2, lines 67-73: The authors should provide more detailed protocols for virus isolation and cell cultures or add the appropriate references.

Response 1: Thank you for pointing it out. We have added the reference to the source of the virus as well as references to the cell culture in lines 68 and 72.

Comment 2: Page 2, lines 84-89: Please define acronyms.

Response 2: Thank you for your suggestion. We have added a table on abbreviations in Materials and Methods (Page 2, lines 90-91) for a clearer and more specific presentation.

Comment 3: Page 2, section 2.4: Please rephrase, this paragraph is very confusing. Is ug used instead of ng?

Response 3: Thank you for your advice. We have reformulated this section (lines 92-97), and clarified the units.

Comment 4: lines 126-127: the PPI network diagram should be included in the results section.

Response 4: Thank you for your suggestion. We have added PPI network diagram in the main manuscript (Figure 4).

Comment 5: Page 9, line 289: Hsp90 is not included in any of the presented tables. Why and how it is commented here?

Response 5: Thank you for your question. Hsp90 has been identified as a protein associated with SHVV replication and although not listed in the table, it was identified as a DEP in both experimental settings in our study (Fold-Change>2 or <0.5, and P-value<0.05). To support the conclusion that glutamine starvation inhibits SHVV replication, we performed an analysis of Hsp90. Thanks for your suggestion, we have listed the Fold-Changes as well as the P-values of Hsp90 (lines 353-355) to make our data and presentation more reliable.

Zhang, Y.; Zhang, Y.A.; Tu, J. Hsp90 Is Required for Snakehead Vesiculovirus Replication via Stabilization of the Viral L Protein. J Virol 2021, 95, e0059421, doi:10.1128/JVI.00594-21.

Comment 6: Page 10, lines 315-317: The proposed mechanism seems interesting, however the authors should provide a more detailed discussion on it, i.e. explain the rationale behind figure 4, possible alternative scenario, etc.

Response 6: Thank you for your suggestion. Figure 4 shows the hypothesis about the mechanism related to SHVV virus replication based on the key signaling pathways and key proteins obtained from our study, combined with the previous study. The study of rationale and the verification of the mechanism may be what we will focus on in the subsequent study.

Furthermore, we adjusted the overall color of some figures to make it more uniform and beautiful..

We have carefully checked our manuscript and revised it according to your suggestion. We hope that our responses and edits are satisfactory. Please contact us if there are any questions. We are looking forward to hearing from you soon.

Thanks again for your hard work!

Best wishes!

Round 2

Reviewer 1 Report

The authors have carefully reviewed the manuscript following the suggestions provided. The manuscript has improved considerably, and the authors have clearly stated in their answers to reviewers the rationale behind their choices for the selected model. It is now ready to be accepted for publication. I congratulate the authors for their work.